# Deciphering triterpenoid saponin biosynthesis by leveraging transcriptome response to methyl jasmonate elicitation in *Saponaria vaccaria*

Xiaoyue Chen[1,2,3], Graham A. Hudson[2,4], Charlotte Mineo[1,2], Bashar Amer[2,5], Edward E. K. Baidoo[2,5], Samantha A. Crowe[2,4,6], Yuzhong Liu [2,4], Jay D. Keasling [2,4,5,6,7,8,9] & Henrik V. Scheller [1,2,3] ✉

Methyl jasmonate (MeJA) is a known elicitor of plant specialized metabolism, including triterpenoid saponins. *Saponaria vaccaria* is an annual herb used in traditional Chinese medicine, containing large quantities of oleanane-type triterpenoid saponins with anticancer properties and structural similarities to the vaccine adjuvant QS-21. Leveraging the MeJA-elicited saponin biosynthesis, we identify multiple enzymes catalyzing the oxidation and glycosylation of triterpenoids in *S. vaccaria*. This exploration is aided by Pacbio full-length transcriptome sequencing and gene expression analysis. A cellulose synthase-like enzyme can not only glucuronidate triterpenoid aglycones but also alter the product profile of a cytochrome P450 monooxygenase via preference for the aldehyde intermediate. Furthermore, the discovery of a UDP-glucose 4,6-dehydratase and a UDP-4-keto-6-deoxy-glucose reductase reveals the biosynthetic pathway for the rare nucleotide sugar UDP-D-fucose, a likely sugar donor for fucosylation of plant natural products. Our work enables the production and optimization of high-value saponins in microorganisms and plants through synthetic biology approaches.

Jasmonates, including jasmonic acid and methyl jasmonate (MeJA), act as ubiquitous and conserved elicitors of plant specialized metabolism through extensive transcriptional reprogramming[1,2]. Saponins are a large group of triterpenoid glycosides found in higher plants and some marine organisms[3]. Their biosynthesis in plants could be triggered by MeJA treatment[3–8]. Despite their structural diversity, saponins share a triterpenoid or steroidal aglycone decorated with sugar moieties[9].

Studies have demonstrated the benefits of saponins to human health[10,11], including anti-inflammatory and anticancer effects, and QS-21, a triterpenoid saponin from *Quillaja saponaria*, is an FDA-approved adjuvant due to its immunostimulatory properties[12–14].

*Saponaria vaccaria* is an annual herb from the Caryophyllaceae family distributed in Asia, Europe, and North America[15]. The seeds of *S. vaccaria* have been used in traditional Chinese medicine (Wang-Bu-

[1]Department of Plant and Microbial Biology, University of California, Berkeley, CA 94720, USA. [2]Joint BioEnergy Institute, 5885 Hollis Street, Emeryville, CA 94608, USA. [3]Environmental Genomics and Systems Biology Division, Lawrence Berkeley National Laboratory, 1 Cyclotron Road, Berkeley, CA 94720, USA. [4]California Institute of Quantitative Biosciences (QB3), University of California, Berkeley, CA 94720, USA. [5]Biological Systems and Engineering Division, Lawrence Berkeley National Laboratory, 1 Cyclotron Road, Berkeley, CA 94720, USA. [6]Department of Chemical & Biomolecular Engineering, University of California, Berkeley, CA 94720, USA. [7]Department of Bioengineering, University of California, Berkeley, CA 94720, USA. [8]Technical University of Denmark, DK-2800 Kongens Lyngby, Denmark. [9]Center for Synthetic Biochemistry, Shenzhen Institutes for Advanced Technologies, Shenzhen, China. ✉e-mail: hscheller@lbl.gov

Liu-Xing, 王不留行) for treating amenorrhea and breast infections[16,17]. Phytochemical studies of seeds and other tissues of *S. vaccaria* have revealed the presence of a large quantity of oleanane-type triterpenoid saponins[18–20]. Their aglycones are derived from β-amyrin. These saponins can be divided into two groups, the monodesmosides containing one oligosaccharide at the C28 of the aglycone, typically gypsogenic acid, and the bisdesmosides with oligosaccharides at C3 and C28 of their aglycones, including quillaic acid and gypsogenin. Although *Quillaja* and *Saponaria* are not closely related phylogenetically, many bisdesmosides in *S. vaccaria* share similar structures to QS-21[15]. In addition, pharmaceutical studies have shown the anticancer

activities of these bisdesmosidic saponins[21,22]. A previous study has reported the discovery of the β-amyrin synthase (βAS) and the C28 triterpene glucosyltransferase involved in the monodesmosides formation in *S. vaccaria*[15]. On the other hand, little is known about the biosynthesis of *S. vaccaria* bisdesmosidic saponins and their relationship to the biosynthesis of monodesmosides.

We used MeJA-elicitation to stimulate saponin biosynthesis in *S. vaccaria*, revealing enzymes for both mono- and bisdesmosidic saponin biosynthesis (Fig. 1). PacBio full-length transcriptome sequencing and gene expression analysis facilitated the discovery of enzymes catalyzing triterpenoid aglycone oxidation and glycosylation.

**Fig. 1 | The biosynthesis pathway leading to the production of monodesmosidic and bisdesmosidic triterpenoid saponins (e.g., Vaccaroside E) in *S. vaccaria*.** Each product of biosynthetic enzyme activity is highlighted in red.

Arrows in solid line indicate reactions confirmed in this study, while arrows in dashed lines indicate hypothetical reactions to be established in the future.

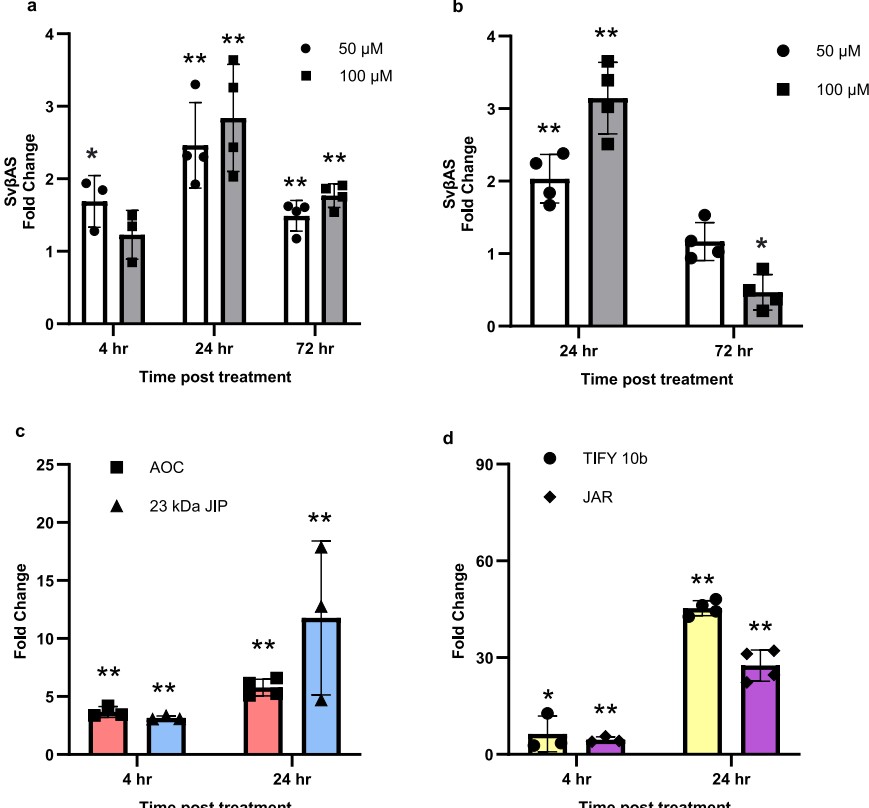

**Fig. 2 | Methyl Jasmonate-elicited transcriptional responses in leaves and flowers of *S. vaccaria*. a** Fold change of *β amyrin synthase* in leaves treated with MeJA at 50, 100 μM compared to 0 μM at 4 h, 24 h, and 72 h tested by qPCR. Error bars indicate mean ± SD (*n* = 3 or *n* = 4 biologically independent samples). **b** Fold change of *β amyrin synthase* treated by MeJA at 50, 100 μM compared to 0 μM at 24 h and 72 h in flowers tested by qPCR. Error bars indicate mean ± SD (*n* = 4 biologically independent samples). **c**, **d** Fold change of *Allene oxide cyclase (AOC)*,

*23 kDa Jasmonate-Induced Protein* (*23 kDa JIP*), *TIFY 10b* and *Jasmonate-Resistant 4* (*JAR*) in leaves treated with 100 μM MeJA at 4 h and 24 h compared to 0 h tested by qPCR. Error bars indicate mean ± SD (*n* = 3 or n = 4 biologically independent samples). Asterisks indicate statistically significant fold change using a one-way ANOVA test with a Tukey HSD test (*$p < 0.05$; **$p < 0.01$). Source data, test statistics, and exact p-values are provided as a Source Data file.

Notably, we found a cellulose synthase-like (Csl) UDP-glucuronosyltransferase that can alter the product specificity of a cytochrome P450 monooxygenase (CYP) towards oxidation of the triterpenoid aglycone, redirecting its involvement from mono- to bis-desmoside biosynthesis. Additionally, we identified a UDP-glucose 4,6-dehydratase and a UDP-4-keto-6-deoxy-glucose reductase for the biosynthesis of the rare nucleotide sugar UDP-ᴅ-fucose, a likely substrate for fucosylation of plant natural products. These findings have implications for enhancing saponin production through metabolic engineering.

## Results

### Methyl jasmonate upregulated expression of *βAS* in leaves and flowers of *S. vaccaria*

Tentative identification of the target mass of vaccaroside E, segetoside I, and segetoside I Ac by LC-MS in different organs of *S. vaccaria* suggested that saponins were present in roots, stems, leaves, and flowers (Supplementary Fig. 1, Supplementary Table 1). Major enzymes involved in the saponin biosynthetic pathway belong to large enzyme families, such as cytochrome P450s (CYPs)[23] and UDP-glycosyltransferases (UGTs)[24,25], making it difficult to distinguish specific enzymes from other members. Therefore, an effective screening method is required for recognizing saponin biosynthesis genes in *S. vaccaria*.

External application of MeJA has been shown to increase saponin production in some plants[3,8,26], often by increasing the expression of saponin biosynthetic genes[4,7]. Therefore, genes involved in the

saponin biosynthetic pathway and the related biological processes could be upregulated together by MeJA, allowing us to narrow down the range of candidate enzymes.

βAS converts 2,3-oxidosqualene into β-amyrin, the first committed step of oleanane-type triterpenoid saponin biosynthesis[27]. MeJA upregulated *S. vaccaria βAS* (*SvβAS*) expression, exhibiting the highest induction after 24 h at 100 μM in both leaves and flowers (Fig. 2). The upregulation of *SvβAS* indicates that exogenous application of MeJA elicited the expression of genes involved in saponin biosynthesis in *S. vaccaria*, potentially leading to elevated saponin production. We confirmed that homologs of other genes that are known to be induced by jasmonates in other plants were similarly upregulated (Fig. 2c, d), including *Allene Oxide Cyclase* (*AOC*)[2], *23* kDa *Jasmonate-Induced Protein* (*23 kDa JIP*)[28], *TIFY 10b*[29] (a JAZ protein), and *Jasmonate-Resistant 4* (*JAR4*)[30].

### Full-length transcriptome sequencing and annotation

We first sought to investigate the complete set of genes co-upregulated with *βAS* by MeJA treatment in *S. vaccaria*, but the complete genome or transcriptome sequences were not available. Therefore, we developed and implemented a pipeline of combinatorial transcriptional sequencing and transcript expression analysis (Supplementary Fig. 2).

To obtain accurate full-length transcriptome sequences from *S. vaccaria*, cDNA libraries from flowers and leaves were constructed for SMRT sequencing by the PacBio Sequel II sequencer[31,32]. A total of 6,104,715 polymerase reads were processed to produce 3,717,290

circular consensus sequencing (CCS) subreads with a mean length of 2388 bp. Next, subreads were refined and clustered, resulting in 118,956 high-quality Iso-seq transcript isoforms from leaves and 113,581 from flowers. After removing redundant transcripts by CD-HIT, non-redundant transcripts from leaves and flowers were combined and collapsed into a total of 89,371 unique transcript isoforms using guidance from the reconstructed coding genome sequences generated by Cogent[33,34]. Cogent partitions Iso-Seq transcripts into gene families based on k-mer similarity, reconstructs the coding region for each gene family, and ultimately creates a de novo coding genome. Subsequently, all Iso-seq transcripts were collapsed into unique isoforms guided by the reconstructed genome. Each Cogent gene family contains unique isoforms, denoted by "PB" with the Arabic number representing the gene it belongs to followed by a period and an Arabic number for the isoform (e.g. 'PB.4332.2'). Isoforms with the same gene number are likely derived from the same gene, but since we do not have a full genome sequence, occasional instances of recently duplicated genes cannot be completely ruled out[34].

The unique transcript isoforms of *S. vaccaria* were annotated by comparing sequences against Swissprot, Pfam, KEGG, and GO databases using BLASTX. As a result, 73,676, 46,788, 64,327, and 71,964 unique transcripts were annotated in the databases mentioned above, respectively. We also detected alternative splicing events in the *S. vaccaria* transcriptome by aligning the nonredundant transcripts to the reconstructed coding genome (Supplementary Fig. 3)[35].

### Illumina sequencing and transcripts profiling

For gene expression profiling, RNA samples from leaves and flowers of *S. vaccaria* with and without MeJA treatment (each in quadruplicates) were subjected to 3'-Tag-RNA-Seq sequencing by Illumina Hiseq. The mapping tool Salmon was then used to map 89,371 unique transcript isoforms obtained from PacBio sequencing for transcript quantification. The Salmon tool tends to map most reads to just one of the isoforms of a gene, typically the isoform with the most complete 3'-end. Importantly, in some cases another isoform may encode a more complete protein sequence and be more suitable for functional characterization. Both principal component analysis (PCA) and hierarchical cluster analysis (HCA) analysis showed that sample replicates of different treatments and tissues correlated well (Supplementary Fig. 4).

### Quantitative reverse transcription-PCR validation of RNA-seq transcript quantification

The reliability of RNA-seq transcript quantification was validated by quantitative reverse transcription-PCR (qRT-PCR). The expression profiles of transcripts obtained from RNA-seq analysis were compared to qRT-PCR results. These are transcripts from the mevalonate and squalene pathways producing triterpenoid precursors: 3-hydroxy-3-methylglutaryl-coenzyme A reductase (*HMGR*) (PB.35046.2), diphosphomevalonate decarboxylase (*MVD*) (PB.5779.2), squalene synthase (PB.40810.6), and βAS (PB.4332.2). Pearson correlation coefficient of log2 value of the expression level fold change is 0.8425, indicating the RNA-seq expression quantification is positively correlated to qRT-PCR results (Supplementary Fig. 5).

### MeJA upregulated saponin biosynthesis pathway genes

To gain insight into the pathways activated by MeJA, we conducted a gene ontology (GO) analysis of differentially expressed genes under MeJA treatment[36]. GO terms associated with both triterpenoid biosynthesis and saponin biosynthesis were significantly enriched among the genes upregulated by MeJA in *S. vaccaria* (Supplementary Fig. 6).

Given that the expression of *SvβAS* and genes involved in squalene synthesis was confirmed to be upregulated by MeJA, it is likely that other saponin biosynthesis genes would respond similarly to this treatment. Therefore, to explore genes co-induced with *SvβAS*, we clustered all the differentially expressed genes based on their

expression patterns. Then we identified a specific subcluster that included *SvβAS* and all other genes that also had increased expression in both leaves and flowers after MeJA treatment (Supplementary Data 1).

### Identification and characterization of SvC28, SvC16, and SvC23 oxidases

We began discovering genes for saponin biosynthesis in *S. vaccaria* with cytochrome P450 monooxygenases (CYPs) involved in the production of triterpenoid aglycones. The aglycone structures of major *S. vaccaria* saponins are oleanane triterpenoids with a C28 carboxylic acid group that could be oxidized at C23 and C16 (Supplementary Table 1): quillaic acid, gypsogenic acid, gypsogenin, etc.

Candidate CYPs for triterpenoid aglycone biosynthesis were identified by gene upregulation and phylogenetic analysis. Several CYPs appeared to be co-induced with βAS upon MeJA treatment in leaves and flowers (Fig. 3a; Supplementary Data 1). Seven CYPs were shown to be co-induced with *SvβAS* and they clustered with known triterpenoid biosynthetic CYPs, suggesting their functions in β-amyrin oxidation.

The previously described C28 oxidases of β-amyrin belong to the CYP716 family[37]. In the neighbor-joining tree, transcript *PB.8389.1* clusters with C28 oxidases from various plants in a subgroup of the CYP716 branch (Fig. 3a). It is also co-upregulated with *SvβAS* by MeJA. To characterize the enzymatic function of the protein encoded by *PB.8389.1*, *Nicotiana benthamiana* leaves were infiltrated with a mixture of *Agrobacterium tumefaciens* strains to express both *SvβAS* and *PB.8389.1*. Subsequent analysis by LC-MS revealed the formation of oleanolic acid, the C28 oxidized β-amyrin, in leaves where both *SvβAS* and *PB.8389.1* were transiently expressed (Fig. 3b, Supplementary Fig. 7). Conversely, oleanonic acid was absent in control leaves infiltrated only with the SvβAS strain. Thus, the function of PB.8389.1 was confirmed to be β-amyrin C28 oxidase in *S. vaccaria* and designated as SvC28 oxidase (CYP716A173).

Two plant CYP enzymes from different subfamilies in the CYP85 clan have been reported to perform C16α oxidation of β-amyrin: *Bupleurum falcatum* CYP716Y1 and *Maesa lanceolata* CYP87D16[38]. In the neighbor-joining tree (Fig. 3a), three *S. vaccaria* transcripts are closely related to MlCYP87D16. However, none of them were induced by MeJA. On the other hand, a subclade was formed by plant CYP716 enzymes, including BfCYP716Y, with a group of potential *S. vaccaria* CYP transcripts. Among these candidates, *PB.2497.1* and *PB.29244.1* exhibited notable co-induction with *SvβAS*. We first selected *PB.2497.1* as candidate C16 oxidase for functional characterization. However, transient expression of *SvβAS*, *PB.8389.1* (SvC28 oxidase), and *PB.2497.1* together in *N. benthamiana* resulted in production of an unknown product with the same *m/z* but with different retention time compared to echinocystic acid, the C16-hydroxylated oleanolic acid. We then selected *PB.29244.4* for functional characterization, as it encodes a full-length protein, whereas *PB.29244.1* has a SNP that causes an early stop codon. Importantly, the product of the protein encoded by *PB.29244.4* in the presence of SvβAS and SvC28 oxidase was echinocystic acid (Fig. 3c, Supplementary Fig. 7).

The function of the protein encoded by *PB.29244.4* was also validated in a β-amyrin-producing yeast strain. Yeast expressing *PB.29244.4* with *SvC28 oxidase* and *Arabidopsis thaliana cytochrome P450 reductase* (*AtATR1*) on plasmid produced echinocystic acid (Supplementary Fig. 8). Thus, PB.29244.4 was identified as the C16α-hydroxylase of β-amyrin in *S. vaccaria* and designated as SvC16 oxidase (CYP716A379).

SvC23 oxidase candidate genes that were co-upregulated with *SvβAS* (PB.6518.3, PB.29444.13, PB.29444.23, and PB.33196.6) resided in two clusters in the tree with C23 oxidases from either CYP714 or CYP72 subfamily (Fig. 3a). *PB.29444.23* and *PB.33196.6* were expressed at a relatively higher level, and their expression fold changes by MeJA were

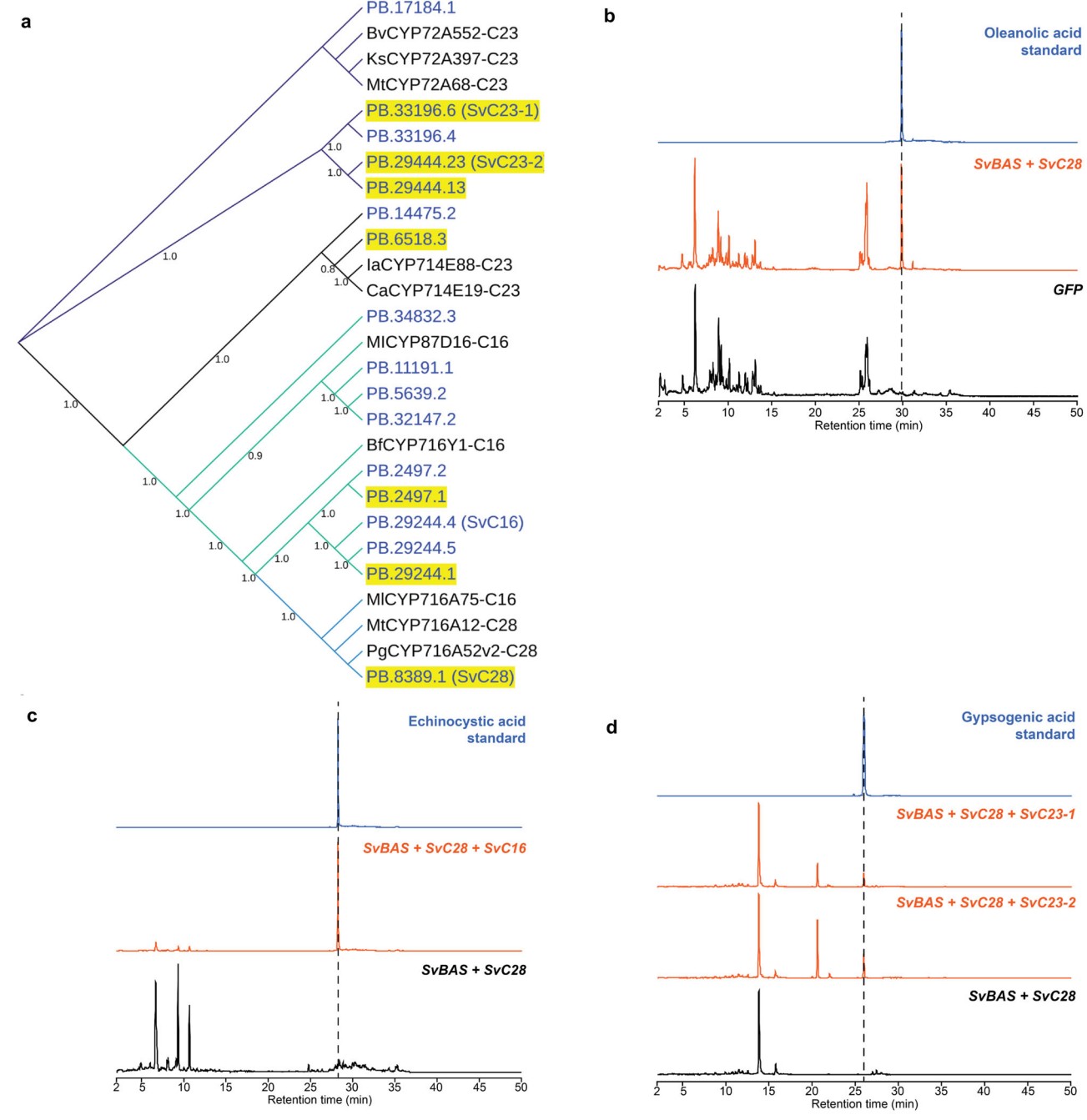

**Fig. 3 | Discovery of genes in the biosynthesis of triterpenoid saponin aglycone in *S. vaccaria*. a** a neighbor-joining tree (1,000 bootstrap replicates) of CYPs acting on triterpenoid from other plants and CYP candidates identified from *S. vaccaria* transcriptome. Gene names in blue represent *S. vaccaria* genes. Yellow highlighted names represent co-upregulated CYPs with β amyrin synthase. Numbers indicate bootstrap values. Sequences of *S. vaccaria* proteins in the tree are in Supplementary Data 2 and accession numbers of CYPs from other species are in

Supplementary Table 2. **b** extracted ion chromatograms (EIC) of oleanolic acid obtained from *Nicotiana benthamiana* transiently expressing *SvβAS + SvC28*. **c** EIC of echinocystic acid obtained from *N. benthamiana* transiently expressing *SvβAS + SvC28 + SvC16*; **d**, EIC of gypsogenic acid obtained from *N. benthamiana* transiently expressing *SvβAS + SvC28 + SvC23-1* and *SvβAS + SvC28 + SvC23-2*. Compounds were identified with authentic standards. Corresponding mass spectra are shown in Supplementary Fig. 7.

the highest among the SvC23 candidates. To investigate the enzymatic functions of these co-upregulated C23 oxidases candidates, we transiently expressed each of them in leaves of *N. benthamiana* with *SvβAS* and *SvC28 oxidase* to provide oleanolic acid as the substrate for C23 oxidases. The C23 oxidation of oleanolic acid can result in three progressively more-oxidized products: hederagenin, gypsogenin, and gypsogenic acid. Neither PB.6518.3 nor PB.29444.13 was able to oxidize oleanolic acid at the C23 position since none of the C23-oxidized oleanolic acid was detected. However, when *PB.33196.6 (SvC23-1*

*(CYP72A1130))* and *PB.29444.23 (SvC23-2 ((CYP72A1131))* were expressed together with *SvβAS* and *SvC28 oxidase*, gypsogenic acid was detected without the presence of hederagenin or gypsogenin (Fig. 3d, Supplementary Fig. 7). The β-amyrin-producing yeast expressing the *S. vaccaria* C23 oxidase-1 or *C23 oxidase-2* in addition to *SvC28 oxidase* and *AtATR1* on plasmid produces exclusively gypsogenic acid, consistent with the results in *N. benthamiana* (Supplementary Fig. 8). Therefore, we identified two C23 oxidases that oxidized oleanolic acid to gypsogenic acid, which is the aglycone of many *S. vaccaria* monodesmosides

(Supplementary Table 1). Thus, these two SvC23 oxidases are likely involved in the biosynthesis of *S. vaccaria* monodesmosides. However, many bisdemosides possess gypsogenin or the C16-hydroxylated gypsogenin (quillaic acid) as their aglycone, and none of these *SvβAS*-co-upregulated SvC23 oxidases or candidates were able to convert oleanolic acid to gypsogenin as a detectable product.

## Combined expression of SvC28, SvC16, and SvC23 oxidases in *N. benthamiana* and yeast

To further investigate the involvement of SvC28, SvC16, and SvC23 oxidases in the biosynthesis of *S. vaccaria* saponin aglycones, we combined their expression in *N. benthamiana* and yeast.

When *SvC28*, *SvC16*, and *SvC23-1/2 oxidases* were expressed with *SvβAS* in *N. benthamiana*, we could not detect echinocystic acid or gypsogenic acid, suggesting that the third oxidase converted them into a new product (Supplementary Fig. 9a, b). We expected that SvC16 oxidase would further oxidize gypsogenic acid, and found evidence for this in several negative ion peaks with $m/z$ 501.4 (Supplementary Fig. 9c). However, we did not detect quillaic acid (QA), the C16α-hydroxylated gypsogenin, in *N. benthamiana*, expressing these three oxidases (Supplementary Fig. 9b). In a β-amyrin-producing yeast strain expressing *SvC28*, *SvC16*, and *SvC23-1/2 oxidases* with *AtATR1* on a plasmid, the later-eluting $m/z$ 501.4 peak was also observed (Supplementary Fig. 10), likely corresponding to the C16α-hydroxylated gypsogenic acid, which is also the aglycone of monodesmosidic segetoside K. This compound is a preferred substrate over gypsogenic acid for a glucosyltransferase to form triterpenoid C28-carboxylic acid glucosides[15]. On the other hand, none of *S. vaccaria* bisdesmosidic saponins have aglycones in the form of C16α-hydroxylated gypsogenic acid (Supplementary Table 1), suggesting its exclusive involvement in monodesmosides biosynthesis.

Unexpectedly, when these three oxidases and *AtATR1* on plasmid were expressed in yeast, the main product was QA (Supplementary Fig. 10), even though combining the expression of *SvC28* and *SvC23-1/2 oxidases* with *AtATR1* in yeast resulted in gypsogenic acid, not gypsogenin. Furthermore, combined expression of these same genes integrated into the yeast genome resulted in a similar product composition (Supplementary Fig. 11). The unforeseen production of QA as the main product demonstrates the enzymatic activity plasticity of both SvC23 oxidases and suggests SvC23-1/2 oxidase may potentially take part in the biosynthesis of bisdesmosides.

Although the product distribution differed in *N. benthamiana* and yeast, we confirmed that aglycones of both mono- and bis-desmosides are formed when *SvC28*, *SvC16*, and *SvC23-1/2 oxidases* are expressed together.

## Identification and characterization of SvCslG

Most bisdesmosidic saponins in *S. vaccaria* and many other species of the Caryophyllales order, have a glucuronic acid residue (GlcA) at the C3 position (Supplementary Table 1). As cellulose synthase-like (Csl) enzymes have been demonstrated to be C3-GlcA transferases of saponins[39–41] and a *Csl* gene was co-upregulated with *SvβAS* (Supplementary Fig. 12), we selected it for functional characterization. We designated the gene *SvCslG* following the nomenclature of Jozwiak and coworkers[39], but as noted by Chung and coworkers[40] the gene is in a clade separate from Arabidopsis *CslGs*.

We first examined whether SvCslG could attach GlcA to the C3 hydroxyl group of QA by incubating microsomal proteins from yeast expressing *SvCslG* with QA and UDP-GlcA. A peak corresponding to QA-C3-GlcA was readily detected in a complete reaction, suggesting that SvCslG transfers the GlcA from UDP-GlcA to C3 of QA (Supplementary Fig. 12).

To further characterize the function of SvCslG, kinetic studies were conducted with various concentrations of QA and UDP-GlcA after establishing the optimal reaction conditions. The substrate

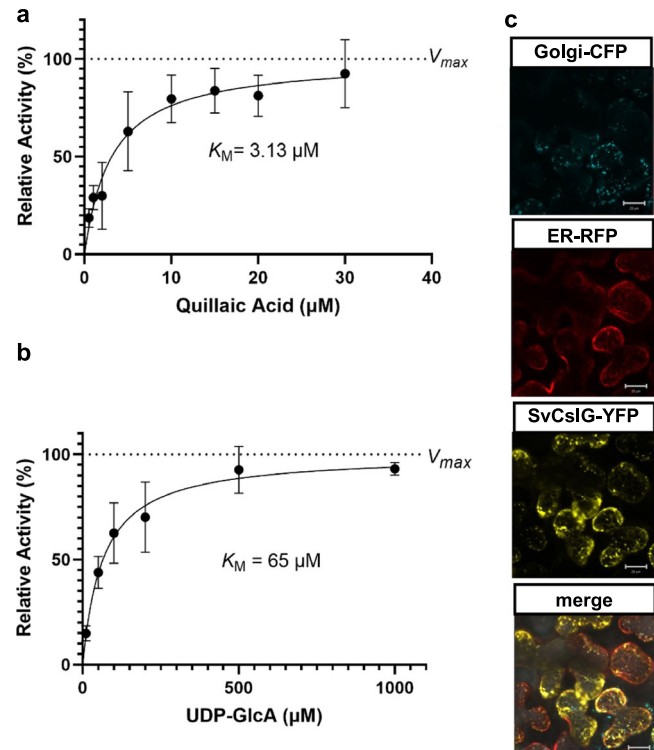

**Fig. 4 | A Cellulose synthase-like (Csl) G from *S. vaccaria* transfers glucuronic acid to C-3 of triterpenoid aglycone quillaic acid.** Kinetic analysis of SvCslG with quillaic acid (**a**) and UDP-GlcA (**b**). Error bars indicate mean ± SD ($n = 3$ biologically independent samples). **c** Confocal images of SvCslG-YFP transiently coexpressed in *N. benthamiana* leaves with Golgi-CFP marker and ER-RFP marker. Scale bars, 20 μm. The experiment was repeated in four biologically independent replicates with similar results. Source data are provided as a Source Data file.

dependency of SvCslG to QA and UDP-GlcA follow Michaelis–Menten kinetics with $K_M$ of 3.13 μM and 65 μM, respectively (Fig. 4).

The function of SvCslG was then tested *in planta* by co-infiltrating leaves of *N. benthamiana* with *A. tumefaciens* carrying *SvCslG* and QA solution. Leaves were collected four days after infiltration and processed to extract saponin components. The peak of QA-C3-GlcA was detected (Fig. 5a, Supplementary Fig. 13a). QA-C3-GlcA was not present in the control samples where SvCslG was replaced with GFP, suggesting that SvCslG utilized the endogenous UDP-GlcA in *N. benthamiana* leaves as a sugar donor and transferred the GlcA to QA, consistent with the in vitro enzymatic assay.

To investigate the subcellular localization of SvCslG, we tagged yellow fluorescent protein (YFP) at the C-terminus of SvCslG and expressed it with ER and Golgi markers in *N. benthamiana*. The overlapped signals of fluorescent protein-tagged SvCslG and organelle markers suggested SvCslG localized at ER and probably also in Golgi (Fig. 4c). Csl enzymes catalyzing C3-glucuronosylation have previously been shown as ER-localized[39,40], unlike most Csl enzymes that are involved in cell wall biosynthesis and localized in Golgi.

## SvCslG facilitates the C3 glucuronidation of C23 aldehyde aglycone

We tested if SvCslG could glucuronidate any triterpenoid aglycone produced by SvβAS, SvC28, SvC16, and SvC23-1/2 oxidases. Although QA was not detected when all three types of β-amyrin oxidases were expressed together in *N. benthamiana*, the additional expression of *SvCslG* gave rise to the formation of QA-C3-GlcA (Fig. 6, Supplementary Fig. 14). In addition, the glucuronide of C16α-hydroxylated gypsogenic acid (GA-C16-OH-C3-GlcA) was also detected by LC-MS (Supplementary Fig. 14), suggesting SvCslG could also glucuronidate GA-C16-OH.

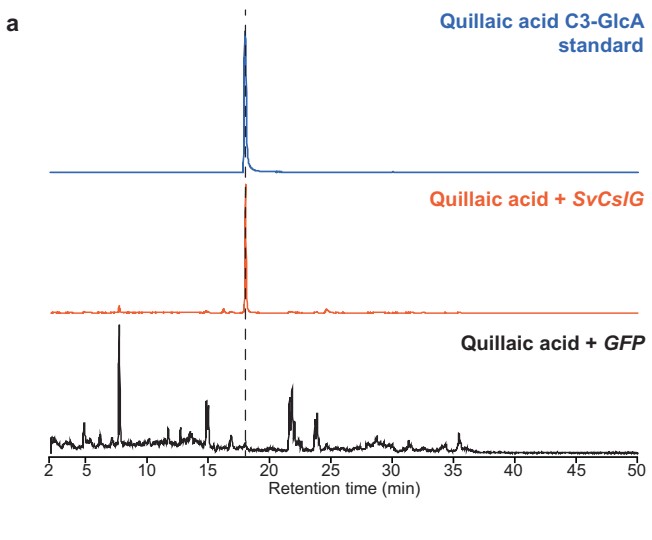

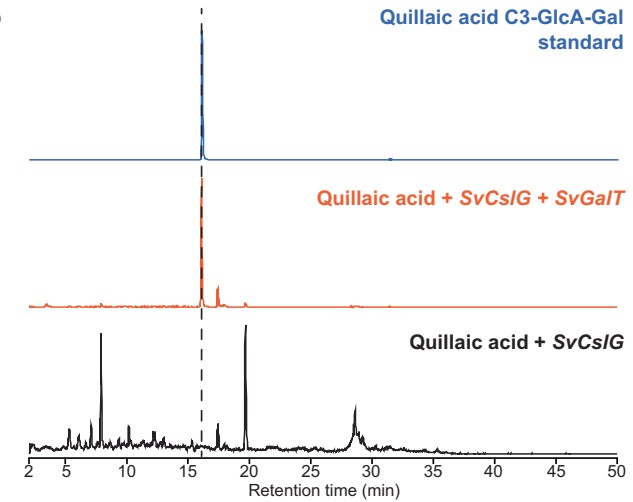

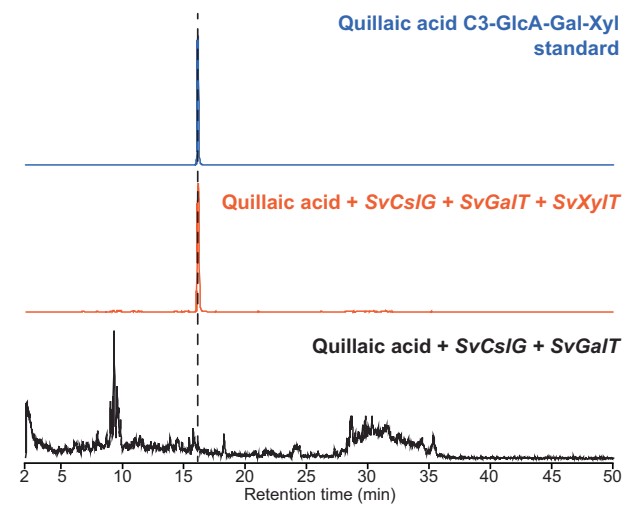

**Fig. 5 | Identification of a galactosyltransferase and a xylosyltransferase glycosylating the 3-O-glucuronide of QA-3-GlcA.** EIC of QA-3-GlcA (**a**), QA-3-GlcA-Gal (**b**), and QA-3-GlcA-Gal-Xyl (**c**) from plants transiently expressing *SvCslG*, *SvCslG* + *SvGalT*, and *SvCslG* + *SvGalT* + *SvC3XylT* as indicated and infiltrated with QA solution. Corresponding mass spectra are shown in Supplementary Fig. 13.

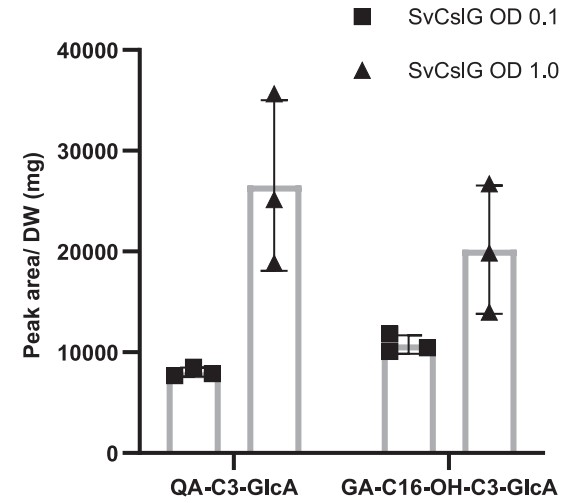

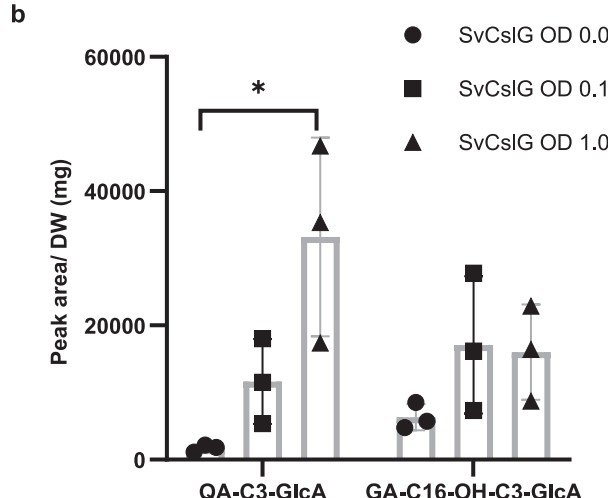

**Fig. 6 | SvCslG increases the production of QA-C3-GlcA.** Different OD levels of *Agrobacterium tumefaciens* harboring *SvCslG* were individually co-infiltrated with *A. tumefaciens* harboring *SvβAS*, *SvC28*, *SvC16* and *SvC23-1* (**a**) and *SvC23-2* (**b**). Error bars indicate mean ± SD ($n = 3$ biologically independent samples). Analysis calculated using a one-way ANOVA test revealed a significant effect of SvCslG strain OD level on the production of QA-C3-GlcA [(**a**), $F(1, 4) = 14.27$, $p = 0.0195$; (**b**), $F(2, 6) = 8.98$, $p = 0.0157$]. Asterisks indicate statistical significance with a Tukey HSD test ((**a**) $p = 0.019$; (**b**) $p = 0.014$). Production of GA-C16-OH-C3-GlcA was not significantly affected by the OD level of SvCslG strain [(**a**), $F(1,4) = 6.39$, $p = 0.0648$; (**b**), $F(2,6) = 1.988$, $p = 0.218$]. Source data and additional $p$-values for Tukey tests are provided as a Source Data file. Compound verification by LC-MS and mass spectra are shown in Supplementary Fig. 14.

The production of QA-C3-GlcA indicates that SvCslG could glucuronidate QA before SvC23 oxidase further oxidizes it into GA-C16-OH. To determine the effect of SvCslG expression on the C23 oxidation of triterpenoids, we co-infiltrated different $OD_{600}$ (optical density) levels of *SvCslG*-carrying *A. tumefaciens* with constant ODs of strains carrying *SvβAS* and all three of β-amyrin oxidases in *N. benthamiana*. As the OD of the SvCslG strain increased, the amount of QA-C3-GlcA increased significantly. At the same time, the production of GA-C16-OH-C3-GlcA did not change significantly (Fig. 6, Supplementary Fig. 14), suggesting that SvCslG facilitated the production of QA as its sugar acceptor and the glycosylation prevented the further C23 oxidation of QA by SvC23 oxidase.

Although gypsogenin was not a detectable product when combining SvβAS, SvC28, SvC16, and SvC23-1/2, we also detected the

formation of gypsogenin-GlcA (GN-GlcA) when SvCslG was expressed. Furthermore, the increasing OD of the *SvCslG*-carrying *A. tumefaciens* was accompanied by a significantly higher concentration of GN-GlcA in *N. benthamiana* leaves after infiltration (Supplementary Fig. 15), indicating SvCslG would also improve the production of gypsogenin and glucuronidate it before another oxidation occurs. These results suggested that SvCslG could partially block the further oxidation of C23 aldehydes by SvC23 oxidases and facilitate their glucuronidation, thus changing the product profile of SvC23 oxidases and redirecting the involvement from the production of exclusive monodesmosides to bisdesmosides.

### Identification of SvGalT and SvC3XylT in C3 glycosylation of QA

A galactose residue is linked to the C3-GlcA residue of many bisdesmosidic saponins in *S. vaccaria*. We constructed a neighbor-joining tree of UDP-glycosyltransferase (UGT) candidates co-upregulated with *SvβAS* in *S. vaccaria* and their homologs with triterpenoid UGTs from other plants (Supplementary Fig. 16). In the neighbor-joining tree, PB.41560.2 was closely related to the triterpenoid-C3-GlcA-Gal transferase (GmUGT73P2) in soybean[42] (Supplementary Fig. 16). For functional verification, it was expressed in the QA-fed yeast with the Arabidopsis UDP-glucose dehydrogenase (*AtUGD*) for UDP-GlcA production and *SvCslG*. However, we could not detect a peak of *m/z* 823.4 corresponding to QA-C3-GlcA-Gal in cell extracts, suggesting PB.41560.2 could not transfer a galactose residue to QA-C3-GlcA. We selected four other *SvβAS*-coinduced UGTs from the same clade of PB.41560.2 (Supplementary Fig. 16) for the galactosyltransferase (GalT) activity test, but none was active.

As the amino acid sequence length of Soyasapogenol-B-GlcA-GalT (495 aa) is longer than all these five candidates, especially compared to PB.41560.2 (460 aa), we hypothesized that PB.41560.2 encodes an incomplete protein. We searched with PB.41560.2 as a query sequence in the transcriptome of *S. vaccaria* to look for a homologous sequence that encodes a full sequence protein. We identified PB.1747.1 as a transcript isoform of PB.41560.2 with only one nucleotide difference at the 3' end of the coding region, suggesting they are two transcript isoforms of the same gene. Although it was not co-upregulated with *SvβAS*, it encoded a longer protein than PB.41560.2 due to the single nucleotide insertion. We deduced that PB.1747.1 would be the best candidate for the C3-GlcA-Gal transferase in *S. vaccaria*.

*PB.1747.1* was transiently expressed with *SvCslG* in *N. benthamiana*, and QA solution was infiltrated with *A. tumefaciens*. A peak with *m/z* 823.4 appeared that was absent in the negative control and matched a QA-C3-GlcA-Gal standard (Fig. 5b, Supplementary Fig. 13b). The function of PB.1747.1 was also validated in yeast using the same method to test PB.42560.2. Therefore, the activity of PB.1747.1 was confirmed as QA-C3-GlcA galactosyltransferase SvGalT (UGT73DL2). The lack of observed activity of PB.41560.2 is likely due to a premature stop codon event caused by the single nucleotide deletion.

QA and gypsogenin 3-*O*-trisaccharide saponins have also been identified in *S. vaccaria* with a xylosyl residue linked to the 3-*O*-glucuronyl group (Supplementary Table 1)[19]. We identified a xylosyltransferase candidate (SvC3XylT (UGT73CC10)) through phylogenetic (Supplementary Fig. 16) and co-upregulation analyses. SvC3XylT resides in the UGT73 family and is related to GmUGT73P2. The activity of SvC3XylT was tested by expressing its gene with *SvCslG* and *SvGalT* in *N. benthamiana* infiltrated with QA solution. The peak of *m/z* 955.4 appeared and was verified as QA-C3-GlcA-Gal-Xyl by the same mass and retention time as the standard (Fig. 5c, Supplementary Fig. 13c).

### SvGalT and SvC3XylT further boost the C3 glycosylation of C23 aldehyde substrates

As shown in the above experiments, although SvCslG improves the biosynthesis of QA-C3-GlcA, GA-C16-OH-C3-GlcA was still produced at approximately 76% of QA-C3-GlcA (Fig. 6). Therefore, we investigated the effect of expressing *SvGalT* and *SvC3XylT* on the proportion of saponins with C23 aldehyde aglycone by comparing the production of C3-glycosylated QA and GA-C16-OH.

By including the expression of *SvGalT*, GA-C16-OH-C3-GlcA-Gal formed at about 22.6% of QA-C3-GlcA-Gal. Furthermore, adding both *SvGalT* and *SvC3XylT* resulted in GA-C16-OH-C3-GlcA-Gal-Xyl produced at only 4.9% of QA-C3-GlcA-Gal-Xyl (Supplementary Fig. 17). Therefore, adding additional sugar moieties at C3 substantially increased the proportion of saponins with QA aglycone. This suggested that SvGalT and SvC3XylT favor substrates with the C23 aldehyde group and boost the formation of C3 glycosylated C23 aldehydes.

### UDP-ᴅ-fucose biosynthesis and C28 fucosyltransferase

Fucose moieties can be found in cell wall polysaccharides and glycoproteins in the ʟ-fucose form derived from GDP-ʟ-fucose. However, the C28 fucose moiety in *S. vaccaria* bisdesmosidic saponins is the β-ᴅ form. Adding UDP-α-ᴅ-fucose to leaf extracts led to the incorporation of the β-ᴅ-fucose molecule into digitoxigenin[43]. Therefore, we hypothesized UDP-α-ᴅ-fucose would provide the fucose moiety in many *S. vaccaria* bisdesmosidic saponins.

The biosynthesis of UDP-α-ᴅ-fucose in plants has not been elucidated previously. However, a dTDP-glucose 4,6-dehydratase and a dTDP-4-keto-6-deoxy-glucose reductase have been reported to convert dTDP-α-glucose into dTDP-α-ᴅ-fucose in *Geobacillus tepidamans*[44]. Thus, we hypothesized that the biosynthesis of UDP-α-ᴅ-fucose would be similar, involving a UDP-glucose 4,6-dehydratase (46DH) and a UDP-4-keto-6-deoxy-glucose (UDP-4K6DG) reductase.

We identified a homolog of the N-terminal domain of *A. thaliana* UDP-rhamnose synthase (AtRHM1) with 46DH activity[45] that was co-upregulated with *SvβAS* in *S. vaccaria* (Supplementary Fig. 18). Furthermore, a homolog of the full-length *AtRHM1* in the *S. vaccaria* transcriptome was also co-upregulated with *SvβAS*. Since the single domain 46DH has not been previously reported in other plants and was induced by MeJA, we hypothesized that Sv46DH converts UDP-Glucose into UDP-4K6DG for the biosynthesis of UDP-ᴅ-fucose. MeJA also induced expression of the putative *SvRHM*, likely involved in UDP-ʟ-rhamnose biosynthesis. On another note, a reductase from the SDR114 family[46] (Supplementary Fig. 19) was selected as the candidate for UDP-4K6DG reductase (SvNMD) as it was co-induced with *SvβAS* and closely related to QsFucSyn, which reduces 4-keto-6-deoxy-glucose to ᴅ-fucose on a saponin backbone[41].

To validate the functions of Sv46DH and SvNMD, we transiently expressed the genes in *N. benthamina* leaves together or individually with *GFP*. When *Sv46DH* and *SvNMD* were combined for expression, both UDP-ʟ-rhamnose and UDP-ᴅ-fucose were produced (Fig. 7a). Overexpression of *Sv46DH* with *GFP* resulted in the production of UDP-ʟ-rhamnose as the predominant product, with only a small amount of UDP-ᴅ-fucose detected. This suggest that the native RHM in *N. benthamina* likely converted the UDP-4K6DG product of Sv46DH into UDP-ʟ-Rhamnose. The small amount of UDP-ᴅ-fucose production was likely due to an endogenous reductase in *N. benthamina* catalyzing the 4-keto reduction of UDP-4K6DG. In contrast, overexpression of *SvNMD* with *GFP* only led to the formation of UDP-ᴅ-fucose, indicating that SvNMD efficiently channeled UDP-4K6DG produced by *N. benthamina* RHM into UDP-ᴅ-fucose. Expressing *GFP* alone in *N. benthamina* leaves did not result in detectable levels of UDP-ʟ-rhamnose or UDP-ᴅ-fucose. UDP-ʟ-rhamnose is expected to be produced in untransformed *N. benthamiana* but did not accumulate at detectable levels. Our results confirmed the involvement of Sv46DH and SvNMD in the biosynthesis of UDP-ᴅ-fucose.

We identified the SvC28FucT candidate (UGT74CD2) based on phylogenetic analysis (Supplementary Fig. 16) and gene expression profile in *S. vaccaria*. There were two GT1-type glycosyltransferase sequences residing in the same subclade with the C28FucT in spinach (SOAP6)[39] and *Q. saponaria*: PB.28124.1 and PB.12216.1 (Supplementary

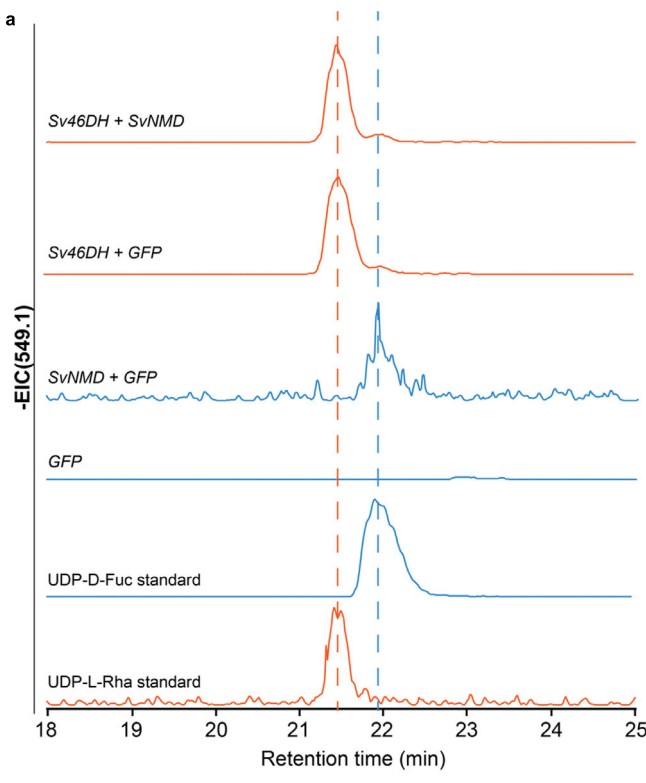

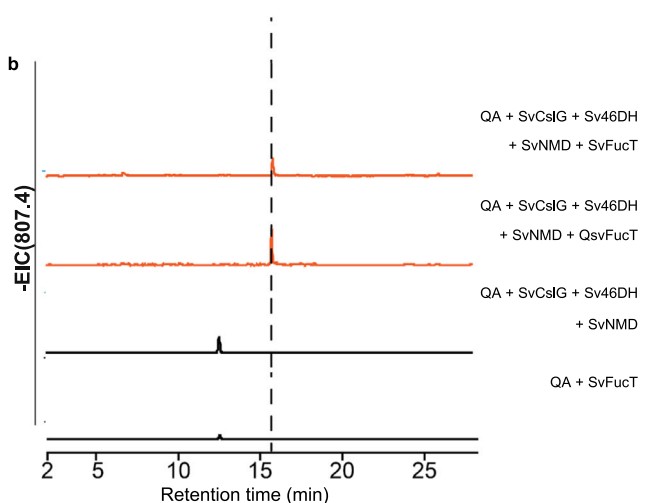

**Fig. 7 | Discovery of the UDP-ᴅ-fucose biosynthetic pathway and a triterpenoid C28 fucosyltransferase in *S. vaccaria*. a** EIC of UDP-ᴅ-fucose and UDP-ᴅ-rhamnose with *m/z* 549.1 from plants transiently expressing *Sv46DH + SvNMD*, *Sv46DH + GFP*, *SvNMD + GFP*, and *GFP*. **b** EIC of GlcA-3-QA-28-Fuc with *m/z* 807.4 from QA-infiltrated plants transiently expressing *SvCslG + Sv46DH + SvNMD+SvFucT*, compared to *SvCslG + Sv46DH + SvNMD+QsFucT* (positive control), *SvCslG + Sv46DH + SvNMD + GFP* (negative control), and *SvCslG+ SvFucT*. Corresponding mass spectrum is shown in Supplementary Fig. 20.

Fig. 16). They are probably a pair of alternative splicing isoforms caused by an intron retention event that converted *PB.12216.1* into *PB.28124.1*. Although they were not identified from the list of genes that were co-upregulated with *SvβAS* in *S. vaccaria*, their nucleotide sequences were very similar to *PB.28124.2*, which was a *SvβAS* co-expressed transcript. Therefore, we chose PB.12216.1 as the SvC28FucT candidate for functional characterization.

To determine if the SvC28FucT candidate could add a fucose moiety to QA-C3-GlcA, we transiently expressed it together with *SvCslG*, *Sv46DH*, and *SvNMD* in *N. benthamina* leaves infiltrated with

QA solution (Fig. 7b, Supplementary Fig. 20). We detected an *m/z* 807.4 peak, corresponding to QA-C3-GlcA-C28-Fuc, which was absent when *SvC28FucT* was replaced with *GFP*, indicating that SvC28FucT was responsible for its formation. The *m/z* 807.4 peak was not detected when both Sv46DH and SvNMD were excluded. When a *C28FucT* from *Q. saponaria*[41] was expressed together with *SvCslG*, *Sv46DH*, and *SvNMD* in *N. benthamiana* leaves injected with QA solution, a peak with the same *m/z* value and retention time was detected. This further confirms that the formation of the *m/z* 807.4 peak requires Sv46DH, SvNMD, and a C28FucT. Based on these results, it is likely that SvC28FucT can transfer the fucosyl residue from UDP-ᴅ-fucose to QA-C3-GlcA. It is also possible that before being reduced to UDP-ᴅ-Fucose, 4K6DG was first attached to the C28 carboxylic group of QA-C3-GlcA by SvC28FucT and then the 4-keto group of 4K6DG linked to the backbone was reduced by SvNMD, giving rise to QA-C3-GlcA-C28-Fuc[41]. Future experiments will be required to resolve whether SvNMD also reduces the 4-keto group of 4K6DG attached to the triterpenoid backbone.

### Functional characterization of other SvβAS co-upregulated glycosyltransferases

The *SvβAS* co-induced GTs PB.29740.3, PB.33723.2, and PB.17537.3 could not add galactose to the glucuronic acid residue of QA-C3-GlcA. We then investigated if they were involved in modifying QA-C3-GlcA-C28-Fuc by combining the expression of each candidate glycosyltransferase with QA-C3-GlcA-C28-Fuc-producing enzymes in *N. benthamiana*. None of these candidates were able to glycosylate QA-C3-GlcA-C28-Fuc. Then we elongated the C28 sugar chain by expressing a C28 rhamnosyltransferase identified from *Q. saponaria*[41] with other QA-C3-GlcA-C28-Fuc producing enzymes, giving rise to a compound with the predicted mass of QA-C3-GlcA-C28-Fuc-Rha (Supplementary Fig. 21). PB.29740.3 could add either a hexose (Hex) or a deoxyhexose (DOH) residue to this substrate, while the other two candidates did not exhibit any activity (Supplementary Fig. 21). Expressing a C28 xylosyltransferase from *Q. saponaria*[41] further elongated the C28 sugar chain to make a product consistent with the predicted mass of QA-C3-GlcA-C28-Fuc-Rha-Xyl (Supplementary Fig. 22). Both PB.29740.3 and PB.17537.3 could glycosylate QA-C3-GlcA-C28-Fuc-Rha-Xyl with a Hex or a DOH; the different retention time of corresponding products suggests they attached Hex or DOH to varying positions of the substrate (Supplementary Fig. 22). Based on known structures of *S. vaccaria* saponins we propose that the hexose and DOH represent glucose and ᴅ-fucose as known to be present in vaccaroside I (Supplementary Table 1). A pentose residue was attached to QA-C3-GlcA-C28-Fuc-Rha-Xyl by expressing PB.33723.2 together with the substrate-making enzymes in *N. benthamiana*. The pentose may be ʟ-arabinofuranose that is attached to ᴅ-fucose in many *S. vaccaria* saponins (Supplementary Table 1, Supplementary Fig. 1). Further experiments are necessary to confirm the structures produced by these transferases.

## Discussion

The elucidation of enzymes catalyzing β-amyrin oxidation and glycosylation support that our strategy utilizing MeJA-elicited transcriptome is useful for mining saponin biosynthesis genes in *S. vaccaria*. It is expected that our transcriptome dataset will benefit the identification of other saponin modifying enzymes and transcriptional regulators of saponin biosynthesis. Meanwhile, this pipeline could also be applied to investigate different MeJA-activated specialized metabolic pathways, including terpenoids, phenylpropanoids, and alkaloids, in any plant without an available genome sequence. The PacBio sequencing enabled us to obtain a large number of accurate and mostly full-length transcripts, which is an advantage over Illumina sequencing, which gives many short reads that are difficult to assemble into full-length transcripts with high confidence. However, the large number of Illumina reads enables much more robust expression

analysis than can be achieved with PacBio sequencing alone. By combining the two methods we were able to both obtain full-length coding regions with high confidence and have the sequencing depth necessary for detecting upregulated transcripts.

Functional characterization of a Csl-type glucuronosyltransferase in *S. vaccaria* showed that Csl enzymes could not only glucuronidate C3 of triterpenoids[39,40] but also affect the activity of P450 β-amyrin oxidase. The ER localization of SvCslG indicates it might change the product profile of C23 oxidase with the help of a close spatial distance between SvCslG and C23 oxidase proteins. The spatial proximity could enable the rapid delivery of the aldehyde intermediates from the active site of C23 oxidase to that of SvCslG before it is further oxidized into the aglycone of monodesmosides. In this way dynamic rearrangements of metabolons could determine if monodesmosides or bidesmosides are predominantly formed[47]. Hence, our finding suggested the necessity of an ER-localized Csl-type enzyme rather than a soluble family 1-type UGT for efficient glucuronidation of particular triterpenoid aglycones. On the other hand, this underlines the importance of the functional plasticity of the C23 oxidase in diversifying the saponin biosynthesis. Furthermore, the increasing ratio of C23 aldehyde triterpenoid C3-glycosides by incorporating other C3 UGTs illustrated that C3GalT and C3XylT were essential in reinforcing the C3 glycosylation of C23 aldehyde aglycones, which is consistent with the structures of aglycones and C3 sugar chains of *S.vaccaria* bisdesmosides. Our findings shed light on how mono- and bis-desmosides' biosynthesis is connected and separated in *S. vaccaria*.

The elucidation of saponin biosynthesis genes in *S. vaccaria* contributes to our understanding of the complex metabolic network of saponins. Furthermore, clarifying the previously unknown biosynthesis pathway of UDP-D-fucose in plants facilitates the reconstitution of fucosylated metabolites, including QS-21, through metabolic engineering in hosts that do not natively produce UDP-D-fucose. In conclusion, our work enables the production and optimization of high-value saponins in microorganisms and plant systems through synthetic biology approaches.

## Methods

### Plant materials and RNA extraction
Seeds of *S. vaccaria* L. 'pink beauty' (johnnyseeds.com) were sown on moist filter paper in the dark at room temperature until germination. Plates of seeds were then placed in a growth room (20 °C, 12 h light) for 3 days. Seedlings were transferred to a growth tray filled with hydroponics solution[48]. Plants were grown in a growth chamber at 50% humidity, 14-h photoperiod, and 18 °C for 5 weeks. Methyl Jasmonate (MeJA) (MillliporeSigma, catalog number 392707) was added to hydroponic solution at 50 or 100 μM. Leaves and flowers were collected from four individual plants of control or MeJA group after 4 to 72 h of treatment. Tissues were immediately frozen in liquid nitrogen and stored at −80 °C until further use.

Total RNA was extracted using TRIzol RNA Isolation Reagent (Thermo Fisher Scientific, Waltham, MA) according to the vendor's manual. DNA was digested and removed with a TURBO DNA-free™ Kit (Thermo Fisher Scientific). RNA was purified and concentrated by using an RNA Clean & Concentrator™ -5 (ZYMO RESEARCH). RNA integrity was measured on a Bioanalyzer 2100 system (Agilent Technologies, Santa Clara, CA). Individual RNA samples with RNA integrity number (RIN) > 7.0 were quantified and aliquoted for library preparation and sequencing as described below.

### PacBio Iso-Seq library preparation and sequencing
Full-length transcript sequencing followed the Isoseq Express Template Preparation protocol from Pacific Biosciences (Menlo Park, CA). Leaf or flower RNA samples were prepared by pooling an equal amount of four biological replicates of both control and MeJA-treated groups. Barcoded double-stranded cDNA libraries were generated using the NEBNext Single Cell/Low Input cDNA Synthesis & Amplification Module, the NEBNext High-Fidelity 2X PCR Master Mix (New England Biolabs, Ipswich, MA) and the Iso-Seq Express Oligo Kit (Pacific Biosciences). SMRTbell sequencing adapters were added to the cDNAs with the SMRTbell Express Template Prep Kit 2.0 (Pacific Biosciences). The libraries were sequenced on a PacBio Sequel *II* sequencer (Pacific Biosciences) at UC Davis Genome Center.

### Illumina 3'-tag-seq library construction and sequencing
Gene expression profiling was conducted using a 3'-Tag-RNA-Seq protocol. Barcoded sequencing libraries were prepared using the Quant-Seq FWD kit (Lexogen, Vienna, Austria) for multiplexed sequencing according to the recommendations of the manufacturer using both the UDI-adapter and UMI Second-Strand Synthesis modules (Lexogen). The fragment size distribution of the libraries was verified via microcapillary gel electrophoresis on a LabChip GX system (PerkinElmer, Waltham, MA). The libraries were quantified by fluorometry on a Qubit fluorometer (LifeTechnologies, Carlsbad, CA), and pooled in equimolar ratios. The library pool was quantified via qPCR with a Kapa Library Quant kit (Kapa Biosystems/Roche, Basel, Switzerland) on a QuantStudio 5 system (Applied Biosystems, Foster City, CA). Up to 48 libraries were sequenced per lane on a HiSeq 4000 sequencer (Illumina, San Diego, CA) with single-end 100 bp reads at UC Davis Genome Center.

### PacBio data processing and detection of AS events
PacBio raw data were analyzed using Iso-Seq v3.2.2 (https://github.com/PacificBiosciences/IsoSeq_SA3nUP) bioinformatics pipeline. First, subreads were processed by CCS v4.2.0 (https://github.com/PacificBiosciences/ccs). Next, cDNA primers and poly-A tails were identified and removed to yield Full-length, nonchimeric (FLNC) transcripts. FLNC transcripts from different tissues were demultiplexed based on barcodes in 5'/3' cDNA primers. Iso-Seq3 was used to refine and cluster FLNC transcripts to obtain transcript consensus with predicted accuracy ≥ 0.99. Redundant transcripts were removed by using CD-HIT-EST with the following parameters: -c 0.99, -n 10, -T 12, -G 0, -aL 0.90, -AL 100, -aS 0.99, -AS 30. The resulting nonredundant transcripts were assembled to reconstruct unique transcript models with Cogent v6.0.0 (https://github.com/Magdoll/Cogent). Then, nonredundant transcripts were mapped to UniTransModels using minimap2 v2.9. Nonredundant transcripts were further collapsed into unique transcript isoforms via cDNA Cupcake v 12.1.0 (https://github.com/Magdoll/cDNA_Cupcake) with parameters as -c 0.95, -i 0.85. Alternative splicing events were detected with Astalavista (http://astalavista.sammeth.net/) using the raw.gtf files from nonredundant transcripts collapsed.

### Transcript functional annotation
Unique transcript isoforms were annotated with Trinotate (https://github.com/Trinotate/Trinotate.github.io/wiki) based on Swissprot[49], KEGG (Kyoto Encyclopedia of Genes and Genomes)[50], GO (Gene Ontology)[51], eggNOG[52] and Pfam[53] database using E-value 10⁻⁵ as a cutoff.

### Illumina data processing and quantification of transcript expression
Raw Illumina reads were cleaned using BBDuk (https://jgi.doe.gov/data-and-tools/bbtools/bb-tools-user-guide/bbduk-guide/) by adapter-trimming, quality-trimming and filtering according to the Quant-Seq data analysis pipeline (https://www.lexogen.com/quantseq-data-analysis/). Then, clean reads were mapped to *S. vaccaria* PacBio full-length unique transcript isoforms using Salmon v1.2.1[54] with the flag −noLengthCorrection for transcript quantification.

### Analysis of differentially expressed transcripts and GO enrichment analysis
Differential expression analysis was performed using limma-voom[55,56]. *P* values were adjusted using the Benjamini-Hochberg procedure[57].

Transcripts that had a $\log_2$ fold change $\geq 0.5$ and adjusted $p < 0.05$ were assigned as differentially expressed. A GO enrichment analysis of differentially expressed genes was performed with the GOseq package[36]. GO terms with an over-represented $p$ value $< 0.05$ were considered as significantly enriched.

## Expression upregulation analysis by clustering

The differentially expressed genes were partitioned into gene clusters with similar expression patterns by cutting the hierarchically clustered gene tree at 60 percent height of the tree with the script "define_clusters_by_cutting_tree.pl" in Trinity (https://github.com/trinityrnaseq/trinityrnaseq/wiki/Trinity-Differential-Expression).

## qRT-PCR validation of expression profile

The same RNA sample was used for both RNA-seq and qRT-PCR. Reverse transcription was performed using qScript cDNA SuperMix (Quantabio, Beverly, MA) according to the manufacturer's instruction. The resulting cDNA was used for qRT-PCR with the BIO-RAD CFX system (BIO-RAD, Hercules, CA). A *GAPDH* gene was chosen as an internal control due to its stable expression under different conditions. Gene-specific primers are listed in Supplementary Data 2. PCR was carried out with three technical replicates with 2X QuantiFast SYBR Green PCR Master Mix (Qiagen, Hilden, Germany) using the following condition: 95 °C for 5 min, followed by 40 cycles of 95 °C for 10 s and 60 °C for 30 s and ended with Melt-Curve analysis. The comparative ΔCt method was used to calculate relative expression levels[58].

## Vector construction and gene cloning for expression in *N. benthamiana*

The gBlocks Gene fragments (Integrated DNA Technologies, San Diego, CA) of CDSs were synthesized for plasmid construction. Except SvCslG and SvC16 oxidases, all other 9 genes were cloned into entry vector Gateway™ pDONR™/Zeo Vector (ThermoFisher) according to the manufacture's guidance. For Gateway cloning, adaptor sequence containing attB1 and attB2 sites were attached at 5′ and 3′ of CDS region. After BP reactions were done, LR recombination reaction were performed with entry vectors harboring each gene and a destination vector by Gateway™ LR Clonase™ Enzyme mix. Destination vectors are listed in Supplementary Data 2. For SvCslG and SvC16 oxidases, CDS were amplified from gBlocks Gene fragments (Supplementary Data 2), and the purified PCR products were Gibson assembled with the linearized pEAQ-HT vector digested with NruI and XhoI. All gene constructs were sequence-verified for downstream applications.

## Transient gene expression in *N. benthamiana*

T-DNA vectors harboring *S. vaccaria* genes were transformed into *A. tumefaciens* GV3101 using electroporation with a Bio-Rad electroporator. Cultures of colonies were grown in LB media containing proper antibiotics for 18 to 22 h before infiltration. The OD$_{600nm}$ of the culture was determined to achieve a final O.D. of 1.0 for each strain if not specified. Different strains were mixed at equal volume for co-infiltration in media (50 mM MES-KOH buffer pH 5.6, 2 mM Na$_3$PO$_4$, 0.5% dextrose, 200 μM acetosyringone (MilliporeSigma, catalog number D134406)). Approximate 1 mL cell suspension was infiltrated into the underside of fully expanded leaves of 4-week-old *N. benthamiana* with 1 mL syringe (no needle). Leaves were harvested four days after infiltration. Three biological replicates consisted of leaves from different plants.

## Metabolite extraction of triterpenoids and saponins in *N. benthamiana*

Leaves were flash frozen immediately after harvest and lyophilized for 2-3 days. Freeze dried leaves were ground twice with Qiagen TissueLyser at 30 Hz for 1 min with 3 mm tungsten beads. Extractions were performed in 80% methanol for 120 min at 25 °C, with shaking at 1400 rpm (Thermomixer Comfort; Eppendorf). Samples were centrifuged at 10,000 $g$ for 5 min and the aqueous phase was entirely dried in speedvac, followed by resuspending in methanol and filtering through 0.22 μm membrane to remove insoluble particles.

## Cloning for expression in *S. cerevisiae*

Coding sequences of *SvCslG* and *SvGalT* were codon optimized for expression in *S. cerevisiae* to synthesize double-strand DNA fragments used as the template for cloning into pESC vectors. The CDS of *SvCslG* was inserted into pESC-URA plasmid using ApaI and SalI restriction enzymes (RE), giving rise to SvCslG-pESC-URA. For Gibson assembly reaction, TTCAACCCTCACTAAAGGGC and TGAATTAACAATTCTTCGCCAGA were attached at 5′ and 3′ of CDS of *AtUGD* when its DNA fragment was synthesized. Then it was assembled with linearized SvCslG-pESC-URA by PacI and NotI RE digestion to make SvCslG-AtUGD-pESC-URA. *SvGalT* and other *SvGalT* candidates were inserted into pESC-LEU plasmid using BamHi and SalI restriction enzymes to generate SvGalT-pESC-LEU.

## Cloning of *S. vaccaria* P450s for functional characterization in yeast

*S. cerevisiae* codon-optimized genes were synthesized by Integrated DNA Technologies. To functionally characterize *S. vaccaria* P450s, plasmids were constructed using Gibson assembly. These plasmids place the genes under galactose-inducible GAL promoters within the pESC-TRP backbone which complements tryptophan auxotrophy in yeast and enables selection in selective "drop-out" media. SvC16 was placed under the SepGAL2 promoter[59] in the sense direction, *A. thaliana Atr1* was placed under pGAL10 in the antisense direction, *SvC28* was placed under pGAL1 in the sense direction, and either isoform of *SvC23* was placed in the antisense direction under pGAL7.

Backbone, gene, or promoter fragments were generated by PCR using Q5 polymerase (New England Biolabs) and amplicons were analyzed and purified by agarose electrophoresis, band excision, and spin column purification using the Zymoclean Gel DNA Recovery Kit (Zymo Research). Gibson assembly was performed using the NEBuilder HiFi DNA Assembly master mix (New England Biolabs) according to the manufacturer's instructions and transformed in to XL1-Blue DH5α and plated on LB agar supplemented with 100 μg/mL carbenicillin (Teknova). Colonies were grown in 5 mL LB media supplemented with 100 μg/mL carbenicillin for 16 h prior to plasmid isolation using the QIAprep Spin Miniprep Kit (Qiagen). Plasmids were sequenced using next-generation sequencing by Primordium Labs and only successfully assembled, error-free plasmids were used in subsequent in vivo experiments in yeast.

## Plasmid-based functional characterization of *S. vaccaria* P450 oxidases

Plasmids were transformed into the β-amyrin-producing strain GHQ14 (genotype: CEN.PK2-1C (erg7::pCTR3-ERG7, erg9::pERG9-ERG9, leu2−3112::His3MX6_PGAL1-ERG19/PGAL10-ERG8, ura3−52::PGAL1-mvaS(A110G)/PGAL10-mvaE(CO), his3Δ1::hphMX4_PGAL1-ERG12/PGAL10-IDI1, YPRCΔ15::pGAL2 ERG1-pGAL10 ERG9-pGAL1-ERG20-pGAL7 GvBAS)) using the lithium acetate method with a 40 min heat shock at 42 °C, plated on CM Glucose Plates minus Tryptophan (Teknova), and incubated at 30 °C for 72 h. Single colonies were used to inoculate culture tubes containing 5 mL of 3X SC -TRP media (5.76 g/L SC -TRP drop out base (Sunrise Science Products), 3.58 g/L yeast nitrogen base lacking amino acids and ammonium sulfate (BD Biosciences), 10 g/L ammonium sulfate (Sigma-Aldrich), buffered to pH 8.0 using mono- and di-basic potassium phosphate) with 2% (w/v) glucose and grown at 30 °C for 48 h. Cultures were aseptically transferred to conical tubes and centrifuged at 3000 × $g$ for 10 min prior to removal of the supernatant. Yeast pellets were then

resuspended in 5 mL of 3X SC -TRP media supplemented with 4% (w/v) galactose and transferred to culture tubes for induction. Cultures were induced for 96 h prior to metabolite extraction and LC-MS analysis as described below.

## Genomic integration and chromosome-based functional characterization of *S. vaccaria* P450 oxidases

Cassettes for overexpression were amplified from successfully assembled and sequence verified plasmids. Amplicons were then integrated into strain GHQ14 targeting the neutral locus ARS1021b using CRISPR-Cas9[60]. Integration was verified by colony PCR and colonies presenting positive hits were transferred to culture tubes containing 5 mL of YPD (10 g/L yeast extract, 20 g/L peptone, 20 g/L dextrose) media and grown at 30 °C for 48 h. Cultures were aseptically transferred to conical tubes and centrifuged at $3000 \times g$ for 10 min prior to removal of the supernatant. Yeast pellets were then resuspended in 5 mL of YPG media (10 g/L yeast extract, 20 g/L peptone, 40 g/L galactose) and transferred to culture tubes for induction. Cultures were induced for 96 h prior to metabolite extraction and LC-MS analysis as described below.

## Metabolite extraction from yeast

1 mL of yeast culture was centrifuged at $21,000 \times g$ for 1 min in screw-top 2 mL plastic vials. The supernatant was removed, and 1 mL of 80% methanol supplemented with 0.1% formic acid was added as well as approximately 100 µL of 0.5-mm glass beads. Yeast was lysed by bead-beating at 30 Hz for 5 min. Lysed yeast was then centrifuged for 5 min at $21,000 \times g$ and the supernatant transferred to fresh 1.7 mL polypropylene centrifuge tubes. The methanolic extract was then evaporated overnight using a LabConco SpeedVac heated to 30 °C. Dried extracts were then reconstituted in 50 µL MeOH by a combination of vortex mixing and sonication. Reconstituted extracts were centrifuged at $21,000 \times g$ for 5 min to remove insoluble debris and the supernatant was analyzed by HPLC-MS.

## Expressing *SvCslG* in yeast microsome protein

Competent yeast (INV*sc*-1) cells transformed with SvCslG-pESC-URA were grown overnight in 15 ml drop-out media at 30 °C and 200 rpm. The seed culture was used to inoculate the main culture (1:20 volume ratio) with 237.5 ml YP media and 0.5% glucose. When $OD_{600} = 0.8$, protein expression was induced with 2% final concentration of galactose. After growing for 12–15 h, cell culture was centrifuged for 10 min at $3000\,g$. Cell pellet was washed with Tris-HCl pH 7.0 and resuspended with disruption buffer (50 mM Tris pH 7.0, 1 mM EDTA, 0.6 M sorbitol, cOmplete™, Mini, EDTA-free Protease Inhibitor Cocktail: Roche). Glass beads (0.425–0.6 mm) were added to the cell suspension in a tube. The tube was shaken at 1/s for 30 s, repeated 20 times with 30-s break on ice. The cell lysate was centrifuged at $15,000\,g$ for 15 min at 4 °C. The supernatants were centrifuged at $50,000\,g$ for 90 min at 4 °C. The microsome pellet was dispersed in buffer (20 mM Tris-HCl pH 7.0, 100 mM NaCl, 20% glycerol).

## SvCslG in vitro enzymatic assay and kinetic analysis

SvCslG enzymatic assay was performed according to Jozwiak et al. 2020[39] with modifications. Each reaction (25 µl) contained 1 µg of microsomal proteins in 20 mM Tris-HCl pH 8.0 buffer with 500 µM UDP-GlcA, 500 µM quillaic acid and 30 mM $Mg^{2+}$ and was incubated at 30 °C. The reaction was quenched with adding 100 µl methanol. Centrifugation at $15,000\,g$ for 15 min at 4 °C was performed and supernatant was then dried to complete using a speedvac. Residues were resuspended in 100% methanol and used for LC-MS analysis. The optimal temperature condition was investigated by incubating reactions for 1 h at different temperatures. To test pH dependency, reactions were incubated in PBS buffer of various pH at 30 °C for 1 h. Enzyme kinetics was calculated using GraphPad Prism software.

## LC-MS detection of triterpenoids in yeast and *N. benthamiana*

Routine LC-MS analysis was performed using an Agilent LC/MSD iQ system with a Kinetex column 2.6 µm XB-C18 100 Å, 100 × 3 mm (Phenomenex). The flow rate was 0.3 mL/min. The binary solvent system was used with solvent A, 0.1% formic acid in water and solvent B, 0.1% formic acid in acetonitrile at gradient: 0-1.5 min, 85% A-15% B; 1.5-26 min, 85%A–15%B to 40%A–60% B; 26–26.5 min, 40%A–60%B to 0% A–100% B; 26.5 min–33 min, 0%A–1000% B; 33-35 min, 0%A–100% B to 85%A–15% B; 35–50 min, 85% A–15% B. Injection volume was 5 µL. The mass detector parameters (ESI-) were set to capillary voltage Negative −3.5 kV and Positive 4.5 kV over a mass range of 420–1450 $m/z$, fragmentor voltage 100 V, 1.0 gain, 150 threshold, 0.10 step size, and scan speed 2600 u/s. Source temperature 325 °C with gas flow 11 L/min and nebulizer 50 psi. Single-ion monitoring (SIM) for quillaic acid and gypsogenic acid was done by monitoring $m/z = 485.4$ with fragmentor voltage 110 V, 1.0 gain, and 290 millisecond dwell time. Analysis was performed using Agilent OpenLab ChemStation software.

## LC-QTOF-MS

For liquid chromatography and high-resolution accurate mass spectrometry, we coupled an Agilent Technologies 1290 Infinity II UHPLC system to an Agilent Technologies 6545 Quadrupole Time-of-Flight Mass Spectrometer (i.e., QTOF-MS). Using the LC gradient profile described above, electrospray ionization (ESI) was achieved with an Agilent Jet Stream ion source. ESI was conducted in the negative ion mode (for measurement of $[M - H]^-$ ions) with capillary and nozzle voltages of 3.5 kV and 2 kV, respectively. Drying and nebulizing gases were set to 10 L/min and 25 lb/in$^2$ (172 kPa), respectively, and a drying-gas temperature of 300 °C was used throughout. The Jet Stream ion source employed a heated nitrogen sheath gas (at 350 °C with a gas flow of 10 L/min) to improve droplet desolvation for signal enhancement. The fragmentor, skimmer, and OCT 1 RF Vpp voltages were set to 140 V, 50 V, and 300 V, respectively. The data acquisition range was from 100-2500 $m/z$, and the acquisition rate was 1 spectra/s. Data acquisition (Workstation B.08.00) and processing (Qualitative Analysis B.06.00) were performed via Agilent Technologies MassHunter software.

## Confocal microscopy analysis

The coding sequence of *SvCslG* in fusion with eYFP was cloned into vector pms057 by Gibson assembly (Supplementary Data 2). Plasmids of endoplasmic reticulum and Golgi marker were obtained from ABRC (ER:CD3-959, Golgi: CD3-961). SvCslG-YFP was transiently expressed together with ER or Golgi marker in *N. benthamiana* epidermal cells. After 3-day post infiltration, leaf disks were collected and analyzed for fluorescence by confocal microscopy using a Zeiss LSM 710 laser-scanning microscope (Carl Zeiss, Oberkochen, Germany) with excitation at 514 nm for YFP, 580 nm for mCherry, and 405 nm for CFP signals.

## Extraction and detection of nucleotide sugars in *N. benthamiana*

Extraction of nucleotide sugars from *N. benthamiana* leaves were performed based on Arrivault et al. 2009[61] with modifications: 2.5 mL chloroform/methanol (3:7, v/v) was added to 100 mg grounded plant tissue immediately. Samples were incubated at −20 °C with mixing every 30 min, followed by extraction with HPLC grade water. The mixtures were centrifuged at $420\,g$ for 10 min. The upper, aqueous-methanol phase was stored at 4 °C. The lower chloroform phase was re-extracted with water, centrifuged as above, and the second aqueous-methanol extract was combined to the first. This step was repeated a third time. The combined aqueous-methanol extract was lyophilized and dissolved in ammonium bicarbonate (5 mM). EnviCarb graphitized carbon columns (Supelclean Envi 57088, Supelco, Pennsylvania, USA) were washed with 60% acetonitrile with 0.30% formic

acid, pH 9.0 and water, followed by loading sample. Then the column was washed with water and 60% acetonitrile. The sugar nucleotides were washed with 60% acetonitrile containing 0.3% formic acid, pH 9.0. Collected samples were lyophilized and dissolved in LC-MS grade water followed by filtering the samples through 0.45 μm Nylon filters.

LC-MS analysis was performed using an Agilent LC/MSD XT system. The column used was a porous graphite carbon column (Hypercarb column, Thermo Scientific, 150 mm × 1 mm, 5 μm). Column was kept at 50 °C during analysis. The flow rate was 0.1 mL/min. The binary solvent system was used with solvent A, 0.3% ammonium formate (pH 9.0) and solvent B, 100% acetonitrile at gradient: 0–20 min, 98%A–2%B to 85%A–15%B; 20–26 min, 85%A–15%B to 50%A–50%B; 26–27 min, 50%A–50%B to 10%A–90%B; 27 min–28.5 min, 10%A–90% B; 28.5–29 min, 10%A– 90%B to 98%A–2%B; 29–45 min, 98%A–2%B. Injection volume was 10 μL. The mass detector parameters (ESI-) were set to capillary voltage Negative −4.0 kV and Positive 4.5 kV over a mass range of 100–1000 $m/z$ with fragmentation voltage 110 V, source temperature 350 °C with gas flow 13 L/min and nebulizer 60 psi. Analysis was performed using Agilent OpenLab ChemStation software.

### LC-MS detection of saponins in *S. vaccaria*
LC was conducted on a Poroshell 120 EC-C18 column (50-mm length, 2.1-mm internal diameter, and 1.9-μm particle size; Agilent Technologies, Santa Clara, CA USA) using an Agilent Technologies 1200 Series Rapid Resolution HPLC system (Agilent Technologies, Santa Clara, CA, USA). A sample injection volume of 2 μL was used. The sample tray and column compartment were set to 6 and 50 °C, respectively. The mobile phases were composed of 0.02% formic acid (Sigma-Aldrich, St. Louis, MO, USA) in water (A) and acetonitrile: water, 2:1, v/v (B). The solvents were of LC-MS grade and purchased from Honeywell Burdick & Jackson, Charlotte, NC, USA. Saponins were separated via gradient elution under the following conditions: linearly increased from 30%B to 37.8% B in 2.5 min, linearly increased from 37.8%B to 90%B in 3.5 min, held at 90%B for 3 min, linearly decreased from 90%B to 30%B in 0.4 min, then held at 30%B for 4 min. The flow rate was held at 0.4 mL/min for 9 min, linearly increased from 0.4 mL/min to 0.6 mL/min in 0.4 min, and held at 0.6 mL/min for 4 min. The total LC run time was 13.4 min. The HPLC system was coupled to an Agilent Technologies 6210 series time-of-flight mass spectrometer (TOF-MS). Drying and nebulizing gas flow rates were set to 11 L/min and 35 lb/in², respectively, and a drying-gas temperature of 320 °C was used throughout. Electrospray ionization was conducted in the positive ion mode with a capillary voltage of 3,500 V for the detection of ions. The fragmentor, skimmer, and OCT 1 RF Vpp voltages were set to 175, 65, and 400 V, respectively. The acquisition range was from 50 to 2200 $m/z$, and the acquisition rate was 1 spectra/sec. Data acquisition and processing were conducted via the Agilent Technologies MassHunter Qualitative Analysis software (B.0.0600).

### Chemical sources
Oleanolic acid was obtained from TCI Chemicals (catalog number 0317). Gypsogenin (catalog number FG74392), gypsogenin C3-GlcA (catalog number MG74054), echinocystic acid (catalog number FE65561), and quillaic acid (catalog number FQ74391)were from Biosynth. Gypsogenic acid was purchased from Santa Cruz Biotechnology (catalog number sc-490170). QA-C3-GlcA, QA-C3-GlcA-Gal, QA-C3-GlcA-Gal-Xyl, and UDP-D-fucose were kindly provided by Dr. Anne Osbourn, John Innes Center[41].

### Phylogenetic analysis
Alignment of protein sequences were performed using ClustalX2. Phylogenetic relationships were inferred using Neighbor-Joining method with 1,000 bootstrap replicates.

### Statistics and reproducibility
We used 16 samples for RNA-seq analyses, with four biological replicates for each sample type. No statistical method was used to pre-determine sample size. After transcript quantification, Principal Component Analysis (PCA) and Hierarchical clustering were conducted to evaluate if biological replicates are well correlated. Data were presented as the mean values ± SD in bar plots. We performed one-way ANOVA test with a Tukey HSD test on the fold change of gene expression by qPCR in Fig. 2 and the quantification of QA-C3-GlcA and GA-C16-OH-C3-GlcA as shown in Fig. 6. No data were excluded from the analyses.

### Reporting summary
Further information on research design is available in the Nature Portfolio Reporting Summary linked to this article.

## Data availability
The unaligned CCS bam files of PacBio sequencing are deposited under NCBI BioProject ID PRJNA951399. Clean and trimmed RNASeq reads are deposited under NCBI BioProject ID PRJNA949801. An annotated list of transcripts with altered expression after MeJA treatment is provided in Supplementary Data 1. The sequences of genes characterized in this work can be found in Supplementary Data 2. Source data are provided with this paper.

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

## Acknowledgements

This project was supported by a grant from GlaxoSmithKline (GSK) to JDK and HVS. We thank the Joint BioEnergy Institute (jbei.org) supported by the U.S. Department of Energy through contract DE-AC02-05CH11231 between Lawrence Berkeley National Laboratory and the U.S. Department of Energy for access to facilities and collaboration. We thank Dr. James Reed and Dr. Laetitia Martin for providing sequence information and cloned genes, Dr. Amr El-Demerdash for providing purified QS-21 pathway compounds, and Dr. Martin Rejzek for providing a standard of UDP-D-fucose. Dr. Jutta Dalton is thanked for help with growing plants. The nomenclature of characterized P450s was kindly reviewed and approved by Dr. David R. Nelson (dnelson@uthsc.edu) and the nomenclature of characterized UGTs was reviewed and assigned by UGT Nomenclature Committee (https://labs.wsu.edu/ugt/).

## Author contributions

X.C., J.D.K., and H.V.S. designed the project. Profiling of saponins in S. vaccaria tissues, X.C., B.A., and E.E.K.B.; plant treatment, RNA purification, transcript quantification by qPCR, bioinformatic analysis (Pacbio data processing, RNA-seq data processing, transcriptome annotation, transcript quantification, coexpression analysis, phylogenetic analysis, gene discovery), X.C.; cloning, screening, and characterizing of candidate enzymes in vitro and in planta, XC; functional verification of P450s in yeast, G.A.H. Functional verification of nucleotide sugar enzymes in yeast, S.A.C. and Y.L.; subcellular localization experiments, C.M. and X.C.; nucleotide sugar profiling, X.C. X.C., G.A.H., and H.V.S. wrote the manuscript with contributions from other authors.

## Competing interests

J.D.K. has financial interests in Amyris, Ansa Biotechnologies, Apertor Pharma, Berkeley Yeast, Cyklos Materials, Demetrix, Lygos, Napigen, ResVita Bio, and Zero Acre Farms. X.C., G.A.H., S.A.C., Y.L., J.D.K., and H.V.S. are inventors of patent applications arising from this work. The remaining authors declare no competing interests.
