## [Peer Review File · Nature Communications]

Deciphering triterpenoid saponin biosynthesis by leveraging transcriptome response to methyl jasmonate elicitation in *Saponaria vaccaria*REVIEWER COMMENTS

Reviewer #1 (Remarks to the Author):

The manuscript reports the elucidation of triterpenoid saponin biosynthesis in *Saponaria vaccaria*, a species of the family of Caryophyllaceae. The authors identified several enzymes, including a cellulose synthase-like one, that decorate the triterpene backbone derived from β -amyrin. The Sv saponins show structural similarities to the soapbark tree saponins, and results from a Caryophyllaceae plant could provide valuable data to understand saponin biosynthesis in different lineages of plants.

However, the work presented in this manuscript seem preliminary composed of raw experimental data, thus are not solid to support their conclusions, including the chemical structure determinations and gene expression analysis.

1. All plots shown in figures (except Figure 1) are in poor quality, and unsuitable for publication.
2. The manuscript is entitled "Deciphering triterpenoid saponin biosynthesis by leveraging transcriptome response to methyl jasmonate elicitation in *Saponaria vaccaria*". To ascertain a successful JA-treatment, expression of mark gene(s) should be analyzed and included in Figure 2.
3. The manuscript provided transcriptome (no genome), but I am not sure whether the transcriptome quality seems not good enough for co-expression analysis and annotation. For example, in line 163 and line 168, the cDNA sequence of the CYP protein gene PB.29244.1 is partial; then PB.29244.4 was claimed to be a complete isoform and used for functional characterization. It is difficult to follow here.

The manuscript seems to confuse the concepts of gene expression upregulation and gene co-expression and, probably, the data provided cannot be used as the basis for gene co-expression analysis.

4. For compound identification, the manuscript only provided EIC plots (without auxiliary lines), even mass spectra were lacking, leading the results unreliable.
5. The manuscript content is incorrectly associated with the picture in many cases, and the gene names are misspelled in many cases. For example, on page 5, lines 157 and 170, the descriptions correspond to Figure 3 rather than Figure 2. On page 6, line 206, the gene name is AtATR1, not AtART1.

Reviewer #2 (Remarks to the Author):

This paper by Chen et al. focuses on elucidating the biosynthesis of triterpenoid saponins in *Saponaria vaccaria*, a plant species that produces a variety of saponins with potential medicinal and industrial applications. The authors used PacBio long-read transcriptome sequencing and MeJA-induced gene co-expression analysis to identify enzymes involved in

the biosynthesis of both mono- and bisdesmosidic saponins, important classes of triterpenoid saponins. They then used biochemical assays to confirm the functions of several candidate enzymes involved in saponin biosynthesis. A novel aspect of this paper is the discovery of a cellulose synthase-like enzyme that not only glucuronidates triterpenoid aglycones but also alters the product profile of a cytochrome P450 monooxygenase via preference for the aldehyde intermediate. The experiments were well designed and executed. The discoveries made enhance our understanding of the biosynthesis of bisdesmosidic saponins. I don't have major concerns. I suggest the authors add some discussion about the use of PacBio long-read transcriptome sequencing in the context of this research and its advantages and disadvantages compared to the more commonly used shotgun approach.

We thank the reviewers for their constructive feedback. We have conducted additional high-resolution LC-MS analyses and in the revised version we provide additional data to confirm the product identifications. We have also added qPCR analysis of marker genes for jasmonate response as requested, and the data confirm that these genes were highly upregulated by the MeJA treatment. The comments of the reviewers have enabled us to make a clearer and better documented presentation of our findings.

Point-by-point response to reviewers' comments:

Reviewer #1 (Remarks to the Author):

The manuscript reports the elucidation of triterpenoid saponin biosynthesis in *Saponaria vaccaria*, a species of the family of Caryophyllaceae. The authors identified several enzymes, including a cellulose synthase-like one, that decorate the triterpene backbone derived from β -amyirin. The Sv saponins show structural similarities to the soapbark tree saponins, and results from a Caryophyllaceae plant could provide valuable data to understand saponin biosynthesis in different lineages of plants.

However, the work presented in this manuscript seem preliminary composed of raw experimental data, thus are not solid to support their conclusions, including the chemical structure determinations and gene expression analysis.

Response:

As detailed below, we have provided additional data to support the findings. We have also provided a more detailed explanation of how the key bioinformatic tools, Cogent and Salmon, work together to generate full-length transcripts and gene expression data for specific genes.

1. All plots shown in figures (except Figure 1) are in poor quality, and unsuitable for publication.

Response:

We had used a lower resolution to keep file sizes small since the journal does not require high resolution figures for initial submission. In the resubmission we have increased the resolution of the uploaded figures. If the paper is accepted, we have high resolution version of all the figures (higher than 300 dpi as required by the journal).

2. The manuscript is entitled "Deciphering triterpenoid saponin biosynthesis by leveraging transcriptome response to methyl jasmonate elicitation in *Saponaria vaccaria*". To ascertain a successful JA-treatment, expression of mark gene(s) should be analyzed and included in Figure 2.

Response:

We have used qPCR to analyze *S. vaccaria* homologs of genes that are known to be induced by JA: *Allene Oxide Cyclase (AOC)*, *23 kDa Jasmonate-Induced Protein (23 kDa JIP)*, *TIFY 10b* and *Jasmonate-Resistant 4 (JAR)*. The data is shown in a new version of Fig. 2. The results show that these other genes are induced as well, in addition to *beta-Amyrin Synthase*. Expression of the marker genes was increased 6- to 45-fold compared to the control at 24 h after induction (even higher than for *beta-Amyrin Synthase*), confirming that MeJA induction worked. It is worth noting that JA-treatment is known to induce some genes transiently and other genes more slowly and persistently. We were specifically interested in genes involved in terpene biosynthesis and used *beta-amyirin synthase* as a marker for this response, and found

24 h to be the optimal time for the induction.

3. The manuscript provided transcriptome (no genome), but I am not sure whether the transcriptome quality seems not good enough for co-expression analysis and annotation. For example, in line 163 and line 168, the cDNA sequence of the CYP protein gene PB.29244.1 is partial; then PB.29244.4 was claimed to be a complete isoform and used for functional characterization. It is difficult to follow here.

The manuscript seems to confuse the concepts of gene expression upregulation and gene co-expression and, probably, the data provided cannot be used as the basis for gene co-expression analysis.

Response:

We agree that 'co-expression' is not the best choice of word as it could suggest that the analysis correlated expression under several different conditions or tissue types. In our study, we only have plus/minus MeJA at 24 h in two organs (flowers and leaves). We have now avoided mentioning 'co-expression' and have instead used terms like 'co-upregulation' and 'co-induced' or simply stated that the expression of genes of interest was increased after MeJA treatment.

We do think that the transcriptome data is a good basis for the analysis, but it was not clear in the original version how the bioinformatics tools work and their limitations. We have added some text to explain this better, while keeping it short to not make the text longer than can be accepted by the journal.

Briefly, Cogent takes individual PacBio Iso-Seq reads and groups them according to their sequence identity into 'genes' and 'isoforms'. Even though PacBio sequencing yields highly accurate sequences ($\approx 99.6\%$ for our study), there can be a few single nucleotide differences between transcripts, and there can also be alternative splicing leading to different isoforms. Isoforms can also represent different allelic forms. Cogent first partitions Iso-Seq transcripts into gene families based on k-mer similarity, then the coding region is reconstructed for each gene family, giving rise to the creation of a *de novo* coding genome using full-length Iso-seq transcripts. Then all the Iso-seq transcripts were collapsed into unique isoforms guided by the reconstructed genes. Each Cogent gene family contains unique isoforms that were used to reconstruct the consensus coding region of a gene. In almost all cases, different isoforms represent transcripts from the same gene, but because there is no reference genome sequence it cannot be excluded that in some cases there could for example be recently duplicated genes with almost identical sequence. PacBio does not give high sequencing depth and is therefore not suitable for gene expression analysis, which was done with Illumina sequencing. Illumina reads were mapped to unique transcript isoforms with the Salmon tool for transcript quantification. For genes with a few highly similar transcript isoforms, it can be difficult for algorithms to correctly assign reads to their true origins. Salmon tends to map most or all reads to the isoform with the most complete 3'-end because the Illumina sequencing is based on 3'-tagged mRNA. However, that isoform is not necessarily the best representative for the protein-coding region. When possible, we have used the isoform that was upregulated according to Salmon for functional analysis since this is the most parsimonious approach. However, in some cases that particular isoform appeared to represent a truncated transcript or had a SNP that causes a premature stop codon. In those cases, we have looked at the alignment of other isoforms for the same gene and characterized another isoform that appears to encode a full-length protein. In the case of gene *PB.29244*, almost all reads were assigned by Salmon to the isoform *PB.29244.1*, which has the most complete 3'-end. However, a SNP in *PB.29244.1* causes a stop codon while *PB.29244.4* encodes a full-length protein. The two other isoforms,

PB.29244.2 and *PB.29244.5* have the same codon as *PB.29244.4* at the position of the SNP giving confidence that this codon is correct and that the SNP in *PB.29244.1* represents a sequencing error.

4. For compound identification, the manuscript only provided EIC plots (without auxiliary lines), even mass spectra were lacking, leading the results unreliable.

Response:

We have reanalyzed samples using high resolution LC-MS with a different instrument. We obtained high resolution accurate mass measurements of analytes in the negative ion mode (for [M – H]⁻ ions) by conducting our LC-MS method on a liquid chromatography quadrupole time-of-flight mass spectrometry system (LC-QTOF-MS). We achieved high mass accuracies (i.e., < 2 ppm) for most of the analytes measured, providing excellent empirical formula determination. As a result, we were able to identify analytes based on their mass spectra and retention times (and compared to bona fide standards where possible). Error values in ppm are given in the plots. New versions are provided of Figures 3, 5, and 7 with supporting new Supplementary Figures 7, 13, and 20 showing the mass spectra with in-source fragmentation and comparisons with standards. New versions are also provided of Supplementary Figures 9, 14, and 15 (corresponding to Supplementary Figures 8, 12, and 13 in the original submission). We have added auxiliary lines where they were lacking to demonstrate congruence in retention times. Some of the compound identifications are putative because we do not have standards, and these cases are explicitly stated. In those cases, we provide the high-resolution mass spectra as Supplementary figures.

5. The manuscript content is incorrectly associated with the picture in many cases, and the gene names are misspelled in many cases. For example, on page 5, lines 157 and 170, the descriptions correspond to Figure 3 rather than Figure 2. On page 6, line 206, the gene name is AtATR1, not AtART1.

Response:

We have carefully checked the manuscript for typographical errors.

Reviewer #2 (Remarks to the Author):

This paper by Chen et al. focuses on elucidating the biosynthesis of triterpenoid saponins in *Saponaria vaccaria*, a plant species that produces a variety of saponins with potential medicinal and industrial applications. The authors used PacBio long-read transcriptome sequencing and MeJA-induced gene co-expression analysis to identify enzymes involved in the biosynthesis of both mono- and bisdesmosidic saponins, important classes of triterpenoid saponins. They then used biochemical assays to confirm the functions of several candidate enzymes involved in saponin biosynthesis. A novel aspect of this paper is the discovery of a cellulose synthase-like enzyme that not only glucuronidates triterpenoid aglycones but also alters the product profile of a cytochrome P450 monooxygenase via preference for the aldehyde intermediate. The experiments were well designed and executed. The discoveries made enhance our understanding of the biosynthesis of bisdesmosidic saponins. I don't have major concerns. I suggest the authors add some discussion about the use of PacBio long-read transcriptome sequencing in the context of this research and its advantages and disadvantages compared to the more commonly used shotgun approach.

Response:

We thank the reviewer for the positive comments. We have added a discussion about the advantages and disadvantages of PacBio sequencing in the Discussion section.

REVIEWERS' COMMENTS

Reviewer #1 (Remarks to the Author):

The manuscript by Chen et al has been improved after revision, presenting valuable information to decipher the biosynthesis of triterpenoid saponins in *Saponaria vaccaria*, a species of the family of Caryophyllaceae, and also an herb used in traditional Chinese medicine. The data of plant treatments, transcriptome analyses and enzyme activity assays are valuable and helpful to further investigation of triterpene biosynthesis in other plants.

There are still some weakness or errors that need improvements or corrections. For example, in Figure 5b, the experiment (middle) and negative control (bottom) are erroneously indicated to contain identical reagents, but the product was absent from "control". For Figure 2e the authors may choose another way (such as heatmap) to present the expression data. I suggest the authors to thoroughly examine their manuscript once again.

Reviewer #2 (Remarks to the Author):

The authors have addressed my minor comment. I have no additional comments.

We thank the reviewers for their feedback on our revised manuscript.

Point-by-point response to reviewers' comments:

Reviewer #1 (Remarks to the Author):

The manuscript by Chen et al has been improved after revision, presenting valuable information to decipher the biosynthesis of triterpenoid saponins in *Saponaria vaccaria*, a species of the family of Caryophyllaceae, and also an herb used in traditional Chinese medicine. The data of plant treatments, transcriptome analyses and enzyme activity assays are valuable and helpful to further investigation of triterpene biosynthesis in other plants.

There are still some weakness or errors that need improvements or corrections. For example, in Figure 5b, the experiment (middle) and negative control (bottom) are erroneously indicated to contain identical reagents, but the product was absent from "control". For Figure 2e the authors may choose another way (such as heatmap) to present the expression data. I suggest the authors to thoroughly examine their manuscript once again.

Response:

We thank the reviewer for finding this error in Fig. 5b. The control was labeled incorrectly. The control contained only quillaic acid and SvCslG, it did not contain SvGalT. This has been corrected.

We agree that Figure 2e was not very informative. A heatmap would not have the necessary resolution to show more than 800 genes. We have decided to remove Fig. 2e and instead describe the subcluster where differentially expressed genes were expressed at higher levels after MeJA treatment in both leaves and flowers.

We have carefully checked the manuscript again for any other errors or inconsistencies.

Reviewer #2 (Remarks to the Author):

The authors have addressed my minor comment. I have no additional comments.

Response:

We thank the reviewer for the comments.